# Evaluation of Internet-Connected Real-Time Remote Auscultation: An Open-Label Randomized Controlled Pilot Trial

**DOI:** 10.3390/jpm12121950

**Published:** 2022-11-24

**Authors:** Takahiro Ito, Takanobu Hirosawa, Yukinori Harada, Shintaro Kakimoto, Taro Shimizu

**Affiliations:** Department of Diagnostic and Generalist Medicine, Dokkyo Medical University Hospital, Tochigi 321-0293, Japan

**Keywords:** telehealth, physical examination, electronic stethoscope, simulator, remote auscultation

## Abstract

The utility of remote auscultation was unknown. This study aimed to evaluate internet-connected real-time remote auscultation using cardiopulmonary simulators. In this open-label randomized controlled trial, the physicians were randomly assigned to the real-time remote auscultation group (intervention group) or the classical auscultation group (control group). After the training session, the participants had to classify the ten cardiopulmonary sounds in random order as the test session. In both sessions, the intervention group auscultated with an internet-connected electronic stethoscope. The control group performed direct auscultation using a classical stethoscope. The total scores for correctly identified normal or abnormal cardiopulmonary sounds were 97/100 (97%) in the intervention group and 98/100 (98%) in the control group with no significant difference between the groups (*p* > 0.99). In cardiac auscultation, the test score in the control group (94%) was superior to that in the intervention group (72%, *p* < 0.05). Valvular diseases were not misclassified as normal sounds in real-time remote cardiac auscultation. The utility of real-time remote cardiopulmonary auscultation using an internet-connected electronic stethoscope was comparable to that of classical auscultation. Classical cardiac auscultation was superior to real-time remote auscultation. However, real-time remote cardiac auscultation is useful for classifying valvular diseases and normal sounds.

## 1. Introduction

Telehealth has emerged as an important aspect of the medical response to the coronavirus disease 2019 (COVID-19) pandemic [1,2,3]. Telehealth can reduce the risk of infection by avoiding unnecessary contact between patients and healthcare workers [4,5]. Remote auscultation is a part of telehealth. Real-time remote auscultation with an electronic stethoscope and an online conferencing system was reported as a very helpful tool [6]. An electronic stethoscope can convert acoustic sounds to electronic signals, which can be further processed and digitalized for transmission to a computer [7]. When the patient is located at a site where the required diagnostic equipment is available, healthcare workers can gather objective measures using an electronic stethoscope and transmit this data to the healthcare providers at a distant site [8].

However, only a few previous studies [9,10,11] have developed and evaluated remote auscultation systems with the physician at a different location. Previous studies in our laboratory have shown that the utility of Bluetooth-connected real-time remote cardiopulmonary auscultation systems is comparable to that of direct auscultation [12]. Utilizing this knowledge, we developed and evaluated the system using an internet-connected electronic stethoscope via an online medical care program to auscultate from more distant sites. The present study aimed to investigate whether physicians remotely auscultated and correctly identified normal or abnormal cardiopulmonary sounds as a screening tool.

## 2. Materials and Methods

### 2.1. Study Design, Setting, and Participants

This study was an open-label randomized controlled trial to evaluate internet-connected real-time remote auscultation using an electronic stethoscope and an online medical care program. We conducted a pilot study because the protocol for real-time remote auscultation using an internet connection has not yet been established. We used a lung simulator for lung auscultation and a cardiology patient simulator for cardiac auscultation to ensure standardized assessment [13]. Simulators also represent a convenient, reliable, and objective method for auscultation skill assessment [14]. The simulators used in this study were located in the skills laboratory at Dokkyo Medical University. In the intervention group, the participants performed real-time remote auscultation at the Doctor’s Office of General Medicine of Dokkyo Medical University. The straight-line distance between the skills lab and the doctor’s office is approximately 220 m. In the control group, the participants performed direct auscultation with a classic stethoscope in the skills laboratory at Dokkyo Medical University.

The study included senior residents and faculty members from the Department of Diagnostic and Generalist Medicine in Dokkyo Medical University Hospital. The hospital was an academic hospital in 2021–2022. The exclusion criteria were refusal to participate in this study or the presence of hearing loss. The study was performed in accordance with the standards of the Declaration of Helsinki. The Institutional Ethics Committee of Dokkyo Medical University, Tochigi, Japan approved the study protocols (No. R-42-16J). The date of approval was 27 January 2021. This study was registered in UMIN-CTR (UMIN000043153; https://upload.umin.ac.jp/cgi-open-bin/ctr/ctr_view.cgi?recptno=R000049259, accessed on 29 January 2021). Written informed consent was obtained from all participants after a detailed explanation of the study before participation.

### 2.2. Procedures

#### 2.2.1. Study Flow and Randomization

The group allocation was conducted using a computer-generated allocation table. Before all sessions, the participants were randomly assigned to either the real-time remote auscultation group (intervention group) or the classical auscultation group (control group). After assignment to each group, the participants performed a training session, followed by a test session.

#### 2.2.2. Training Session

In the intervention group, participants performed real-time remote auscultation using an electronic stethoscope and an internet-connected online medical care program. In the control group, participants auscultated all sounds directly using a classical stethoscope (Littmann Cardiology III; 3M, St Paul, MN, USA), placing it on the lung simulator and the cardiac patient simulator by themselves. The participant listened to five different lung sounds and five different cardiac sounds in a previously determined order, with the correct classification provided for each sound. In the training session for lung auscultation, the following five sounds were played: normal lung sounds, wheezes, rhonchi, stridor, coarse crackles, and fine crackles. In the training session for cardiac auscultation, the following five sounds were played: normal cardiac sounds, third cardiac sound (S3 gallop enhanced), aortic stenosis, aortic regurgitation, and mitral regurgitation. Each participant was instructed to auscultate the simulators at standardized positions: two times each on the anterior (Figure 1a) and posterior sides (Figure 1b) of the lung simulator, and four times on the cardiology patient simulator (Figure 1c). Each sound was played for a maximum of one minute.

#### 2.2.3. Test Session

In the test session, the researchers changed the settings of the cardiology patient simulator to ensure that the monitoring screen of the cardiology patient simulator displayed only a heartbeat icon. In this session, all participants auscultated the five different lung sounds and five different heart sounds from the training session in a random order and filled the types of sounds they recognized in a formatted questionnaire (Appendix A).

#### 2.2.4. Simulator

For lung auscultation, the same lung simulator (MW28; Kyoto Kagaku Co., Ltd., Kyoto, Japan) was used in all sessions. This simulator was designed for medical education training using 34 samples of lung sounds, which were classified as continuous (wheezes or rhonchi) or discontinuous (fine or coarse crackles), according to the classification of the American Thoracic Society [15]. In the lung simulator, the light-emitting diode panel on the simulator side indicates the inspiration or expiration phase.

For cardiac auscultation, the same cardiology patient simulator (MW41; Kyoto Kagaku Co., Ltd., Kyoto, Japan) was used in all sessions. This simulator was designed for medical education training and includes 88 cardiac sounds recorded from actual patients and reproduced using a high-quality sound system. Cardiology patient simulators can present data for vital signs (heartbeat, blood pressure, respiratory rate, and body temperature), electrocardiogram (ECG), carotid artery pulse, jugular vein pulse, and apex cardiogram. In the test session for the control group, the cardiology patient simulator’s monitoring screen was modified to display only a heartbeat icon. In the intervention group, the participants were not allowed to see the monitor.

#### 2.2.5. Real-Time Remote Auscultation

Participants auscultated all sounds remotely using an electronic stethoscope (JPES-01; MEMS CORE Co., Ltd., Miyagi, Japan), a wireless module for the electronic stethoscope option (BioCMOS Co., Ltd., Tokyo, Japan), a noise-canceling stereo headset (WH-1000XM3; Sony Corp., Tokyo, Japan), and an online medical care program (Smart Cure, Smart Gate Inc., Tokyo, Japan) (Figure 2). Researchers placed the electronic stethoscope on the simulator (Figure 3a), and participants could monitor the placement of the electronic stethoscope in real time through the online medical care program (Figure 3b).

The electronic stethoscope is equipped with an ultra-sensitive piezoelectric sensor film and an electrical amplifier, and the signals are converted into sound waves. It is also equipped with a volume regulator and a frequency filter to enable high-quality hearing. The filter has the following modes: a bell mode (20–100 Hz), diaphragm mode (200–2000 Hz), and a wide mode (20–2000 Hz). In the lung and cardiac parts, we used the diaphragm mode (200–2000 Hz) and bell mode (20–100 Hz), respectively. The transmitter transferred the lung and cardiac sounds to the wireless module for the electronic stethoscope option (BioCMOS Co., Ltd.) and the online medical care program via an internet connection.

The online medical care program supports full high-definition videos and 4 K still images, incorporating an online conferencing system (Cisco Webex Meetings, Cisco Systems G.K.). The online medical care program can easily share medical record information, such as medical interview sheets, sounds, and images with doctors. Therefore, in real-time auscultation, multiple doctors can simultaneously hear, record, and share data.

### 2.3. Data Collection and Outcome Measures

We collected data on age, sex, and years since obtaining a degree in medicine from all participants as baseline demographic data. All participants’ answers for each sound in the test session were collected. The primary outcome measure was the total scores for correctly identified normal or abnormal cardiopulmonary sounds of each group. The secondary outcome measures were the rates of correct answers for lung sounds and cardiac sounds.

### 2.4. Statistical Analysis

Results were analyzed with R 4.0.5 for Windows (The R Foundation for Statistical Computing, Vienna, Austria). A post-hoc power analysis was performed using the program G*power 3.1.9.6 [16]. Statistical significance was set at *p* < 0.05. The correct answers in each group were compared using Fisher’s exact test for primary and secondary outcome measures. The Mann–Whitney U test was used to compare the continuous variables for baseline participant characteristics, which were presented as mean and standard deviation (SD). In contrast, Fisher’s exact test was used to compare the categorical and binary variables for baseline participant characteristics, which were presented as numbers (percentages).

## 3. Results

### 3.1. Participants’ Profiles

Twenty physicians in the Department of Diagnostic and General Medicine at Dokkyo Medical University participated in this study (Figure 4). The mean age of all participants was 32 ± 4.7 (standard deviation, SD) years; 14 participants (70%) were males, and the mean number of years since graduation was 7.5 ± 4.8. Ten participants each were assigned to the intervention and control groups. There were no significant intergroup differences in participant age (*p* = 0.62), sex (*p* = 0.14), and years since graduation (*p* = 0.54) (Table 1).

### 3.2. Diagnostic Performance

The scores for correctly identified cardiopulmonary sounds are presented in Table 2. The total scores for correctly identified normal or abnormal cardiopulmonary sounds were 97/100 (97%) in the intervention group and 98/100 (98%) in the control group with no significant difference between the groups (*p* > 0.99).

#### 3.2.1. Diagnostic Performance for Lung Sounds

In lung sounds, the test score was 43/50 (86%) in the intervention group and 45/50 (90%) in the control group, with no significant differences between the groups (*p* = 0.54). The exploratory analysis showed that the two groups also showed no significant differences for identification of normal lung sounds, wheezes, rhonchi, coarse crackles, or fine crackles. The participants in the intervention group did not assign abnormal lung sounds to normal lung sounds. The participants’ responses during lung auscultation are described in more detail in Table 3.

#### 3.2.2. Diagnostic Performance for Cardiac Sounds

In cardiac sounds, the test score was 36/50 (72%) in the intervention group and 47/50 (94%) in the control group, with a significant difference between groups (*p* < 0.05). The exploratory analysis showed that only 5/10 (50%) participants in the intervention group correctly auscultated aortic stenosis, compared with 9/10 (90%) in the control group (*p* = 0.07). In the intervention group, 7/10 (70%) participants correctly auscultated aortic regurgitation and mitral regurgitation, whereas 9/10 (90%) in the control group did so (*p* = 0.28). Participants in both groups showed no misclassifications between normal cardiac sounds and the sounds of valvular diseases (aortic stenosis, aortic regurgitation, and mitral regurgitation). In the intervention group, aortic stenosis was assigned as aortic regurgitation by 3/10 (30%) participants and as mitral regurgitation by 2/10 (20%) participants. On the other hand, aortic regurgitation was assigned as aortic stenosis by 2/10 (20%) participants, and mitral regurgitation was assigned as aortic stenosis by 3/10 (30%) participants. Details of the participant responses in cardiac auscultation are listed in Table 4.

Post-hoc power analysis revealed that the statistical power for comparison of the total test score was 0.09. The statistical power for comparison of the test score in the lung part was 0.2 between the intervention group and the control group. However, the corresponding value in the cardiac part exceeded 0.99.

## 4. Discussion

### 4.1. Principal Results

This study yielded several main findings. First, we have scientifically demonstrated that physicians could remotely identify normal or abnormal cardiopulmonary sounds in more than 95% of cases. This suggested that physicians could remotely auscultate cardiopulmonary sounds as a screening tool. In telemedicine using the optimal methods, physicians could remotely perform physical examination (not only inspection but also auscultation).

The second major finding was that remote pulmonary auscultation was also comparable to direct auscultation (86% vs. 90%, *p* = 0.54). Coarse and fine crackles are helpful in diagnosing pneumonia [17]. The results suggest that real-time remote auscultation is helpful in the diagnosis of pneumonia. The ability to detect this finding by remote auscultation will have a significant impact on clinical practice in the context of the global spread of COVID-19, since in an outpatient department, during ward rounds, or in a doctor’s visit at home for febrile patients, medical staff can screen the patients’ lung sounds without contact and protect themselves from infectious diseases including COVID-19.

The third major finding was that in cardiac auscultation, classical auscultation was still superior to real-time remote auscultation using an internet-connected electronic stethoscope (72% vs. 94%, *p* < 0.05). This study revealed that half of the aortic stenosis cases were misclassified as other valvular diseases in real-time remote cardiac auscultation. In Bluetooth-connected real-time remote auscultation [12], aortic stenosis was correctly identified in 5/6 cases (83.3%), compared with 5/10 (50%) in this study. This finding suggests that real-time remote cardiac auscultation cannot easily classify the type of valvular disease. This could be attributed to the differences in sound quality caused by the internet connection compared with that provided by the Bluetooth connection. This could also be attributed to the difficulties in detecting the systole and diastole phases without palpating pulses.

However, valvular diseases were not misclassified as normal cardiac sounds in real-time remote auscultation. Moreover, normal cardiac sounds were not misclassified as valvular diseases. These results suggest that real-time remote cardiac auscultation may still be useful in identifying whether a cardiac sound is normal or related to a valvular disease, indicating the potential of this technique in facilitating remote screening. In the future, it will be essential to develop higher-quality internet connections or additional support tools to improve real-time remote cardiac auscultation.

### 4.2. Strengths

This study had three major strengths. First, this study demonstrated the utility of an internet-connected real-time remote lung auscultation system. Real-time remote cardiac auscultation may help distinguish between normal sounds and the sounds of valvular diseases. Second, all the physicians who participated in the study were general internal medicine physicians, who routinely use stethoscopes to screen for abnormal auscultation findings [18]. Third, the survey items in this study evaluated five common lung sounds and five common cardiac sounds that are often encountered in actual clinical practice [19,20].

### 4.3. Limitations

This was a pilot study with several limitations. First, the sample size was small and did not include data from actual patients. Fully powered trials are required to show equivalence or non-inferiority, especially for the lung auscultation part. Therefore, future studies with larger sample sizes are required to confirm the utility and validation of real-time auscultation in real clinical settings. Second, the responses may have shown some within-participant dependencies. Third, participants in the intervention group could not decide on the timing of the changes in the auscultation sites, whereas the participants in the control group could change the auscultation sites by themselves, making it easier to detect the systole and diastole phases in classical cardiac auscultation. Fourth, the electronic stethoscope made the noises more pronounced through the internet connection. This characteristic affected the classification of cardiac valvular diseases in the intervention group.

### 4.4. Comparions with Our Prior Work

Our laboratory has already conducted a study directly comparing real-time remote auscultation using a Bluetooth connection with classical auscultation, demonstrating the utility of real-time lung and cardiac auscultation using a Bluetooth connection [12]. These results indicate the effectiveness of real-time remote lung auscultation using both Bluetooth and internet connections. In our previous study, the total cardiac sound score was 50/60 (83.3%) in the Bluetooth-connected real-time remote auscultation group and 119/140 (85.0%) in the control group, with no difference between the two groups. The results obtained with real-time remote cardiac auscultation using an internet connection were poorer than those obtained with remote auscultation based on the Bluetooth connection. This could be because the participants could easily distinguish between inspiratory and expiratory phases in normal lung sounds even in real-time lung auscultation. However, it would be difficult to distinguish between the systolic and diastolic phases in real-time cardiac auscultation only from the quality of cardiac sounds [21].

### 4.5. Other Remote Auscultation Systems

To the best of our knowledge, several remote auscultation systems using electronic stethoscopes, in addition to the one which we have used, are currently available [10,22,23,24,25]. However, almost all studies evaluated only the cardiac sounds [21,22,23,24] or pulmonary sounds [26], not both. Additionally, a few studies directly compare remote auscultation and classical auscultation [11,27,28,29]. The sensitivity and specificity ranged from 87–100% and 82–98%, respectively.

## 5. Conclusions

This study demonstrated that the utility of an internet-connected real-time remote cardiopulmonary auscultation system was comparable to that of classical auscultation. The internet-connected real-time remote cardiopulmonary auscultation system was useful to detect normal or abnormal cardiopulmonary sounds.

The internet-connected real-time remote cardiac auscultation system could distinguish between normal sounds and the sounds of valvular diseases. However, additional tools should be developed to improve real-time remote cardiac auscultation in future studies. For example, a phonocardiogram or real-time ECG can visualize the systole and diastole phases of sounds [30]. Artificial intelligence can also be used to automatically support clinicians in diagnosing cardiopulmonary diseases [31].

The findings of this study can thus pave the way for future studies of real patients and facilitate personalized telehealth.

## Figures and Tables

**Figure 1 jpm-12-01950-f001:**
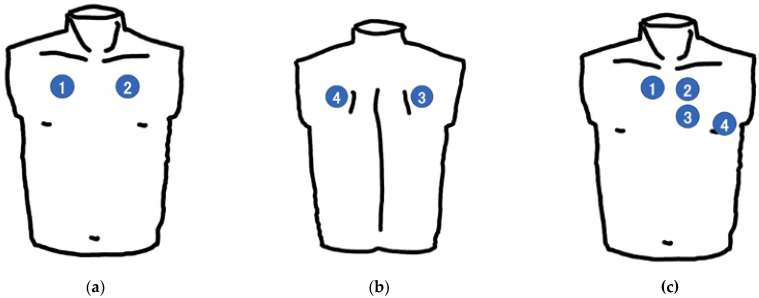
The four different areas of auscultation: (**a**) the anterior of the lung simulator, (**b**) the posterior of the lung simulator, (**c**) the cardiology patient simulator.

**Figure 2 jpm-12-01950-f002:**
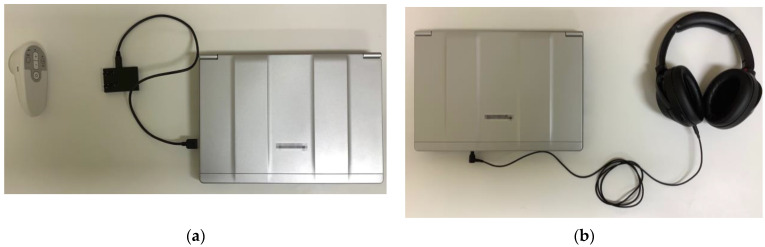
(**a**) Remote auscultation equipment at the researcher site and (**b**) the participant site, including an electronic stethoscope, a wireless module for the electronic stethoscope option, and a noise-canceling stereo headset.

**Figure 3 jpm-12-01950-f003:**
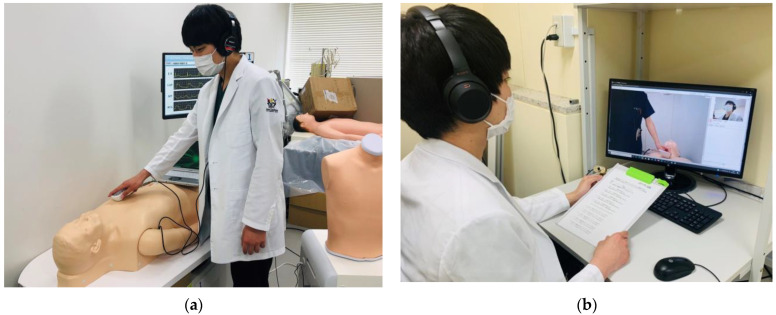
The internet-connected real-time remote auscultation processes. (**a**) The researcher puts an electronic stethoscope on the simulator. (**b**) The participant remotely auscultates the sound with an internet-connected medical system.

**Figure 4 jpm-12-01950-f004:**
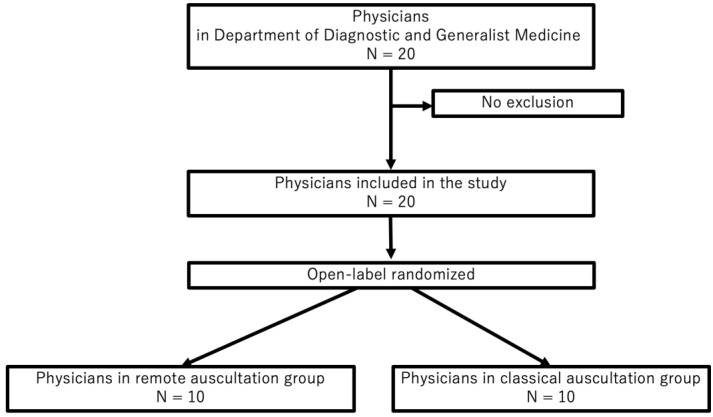
Flowchart of participants in the study.

**Table 1 jpm-12-01950-t001:** Baseline characteristics of participants in the classical and remote auscultation groups.

Variable	Remote Auscultation (N = 10)	Classical Auscultation (N = 10)	*p* Value
Age (years), mean ± SD	31.6 ± 4	32.6 ± 5.1	0.63 ^1^
No. of men, *n* (%)	9 (90)	6 (60)	0.15 ^2^
Years after graduation (years), mean ± (SD)	6.8 ± 4	8.1 ± 5.5	0.54 ^1^

^1^ Mann–Whitney U test; ^2^ Fisher exact test; SD, standard deviation.

**Table 2 jpm-12-01950-t002:** Correct identification rates of sounds.

Variable	Remote Auscultation (N = 10)	Classical Auscultation (N = 10)	*p* Value ^1^
Total scores for correctly identified normal or abnormal cardiopulmonary sounds	97/100 (97)	98/100 (98)	>0.99
Total lung sounds, *n* (%)	43/50 (86)	45/50 (90)	0.54
Normal	10/10 (100)	10/10 (100)	>0.99
Wheeze	10/10 (100)	9/10 (90)	0.99
Rhonchi	9/10 (90)	9/10 (90)	>0.99
Coarse crackles	8/10 (80)	9/10 (90)	0.54
Fine crackles	6/10 (60)	8/10 (80)	0.34
Total cardiac sounds, *n* (%)	36/50 (72)	47/50 (94)	<0.05
Normal	9/10 (90)	10/10 (100)	0.99
S3 gallop	8/10 (80)	10/10 (100)	0.99
Aortic stenosis	5/10 (50)	9/10 (90)	0.07
Aortic regurgitation	7/10 (70)	9/10 (90)	0.28
Mitral regurgitation	7/10 (70)	9/10 (90)	0.28

^1^ Fisher exact test.

**Table 3 jpm-12-01950-t003:** Detailed answers in lung auscultation.

	Participants’ Responses in the Remote Auscultation Group				
	Normal	Wheeze	Rhonchi	Coarse Crackles	Fine Crackles
Correct Answer					
Normal	10/10	0	0	0	0
Wheeze	0	10/10	0	0	0
Rhonchi	0	0	9/10	0	1/10
Coarse crackles	0	0	0	8/10	2/10
Fine crackles	0	0	1/10	3/10	6/10
	Participants’ Responses in the Classical Auscultation Group				
	Normal	Wheeze	Rhonchi	Coarse Crackles	Fine Crackles
Correct Answer					
Normal	10/10	0	0	0	0
Wheezes	0	9/10	1/10	0	0
Rhonchi	0	1/10	9/10	0	0
Coarse crackles	0	0	0	9/10	1/10
Fine crackles	1/10	0	0	1/10	8/10

**Table 4 jpm-12-01950-t004:** Detailed answers in cardiac auscultation.

	Participants’ Responses in the Remote Auscultation Group				
	Normal	S3 gallop	Aortic stenosis	Aortic regurgitation	Mitral Regurgitation
Correct Answer					
Normal	9/10	1/10	0	0	0
S3 gallop	2/10	8/10	0	0	0
Aortic stenosis	0	0	5/10	3/10	2/10
Aortic regurgitation	0	1/10	2/10	7/10	0
Mitral regurgitation	0	0	3/10	0	7/10
	Participants’ Responses in the Classical Auscultation Group				
	Normal	S3 Gallop	Aortic Stenosis	Aortic Regurgitation	Mitral Regurgitation
Correct Answer					
Normal	10/10	0	0	0	0
S3 gallop	0	10/10	0	0	0
Aortic stenosis	0	0	9/10	1/10	0
Aortic regurgitation	0	0	1/10	9/10	0
Mitral regurgitation	0	0	1/10	0	9/10

## Data Availability

Not applicable.

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
