# Peer review of "Evaluation of Internet-Connected Real-Time Remote Auscultation: An Open-Label Randomized Controlled Pilot Trial"

_jpm, 2022, doi:10.3390/jpm12121950_

Round 1
Reviewer 1 Report
The paper is well written and documented. The organization is good. The presentation is good. Readability is good. References are regular. Although more detailed considerations on performance would improve the value of the paper, it seems to me suitable to be accepted for publication in this journal.
As noted in the comments below, there are a few issues that would benefit from clarification or further discussion before the paper is accepted:
1) Could you please rewrite the introduction? This section, in my opinion, is confusing to readers. I'd like to highlight a few key points: background, problem, proposed solution, and main contributions Furthermore, this would help to reinforce the topic's importance in the telehealth sector.
2) It is essential to include a separate section for the literature review, with the goal of comparing this work to others in the field and highlighting what makes it unique. It would also help to expand the references.
3) The author should explain how to consider the interaction model to ensure the robustness of the architecture. Specifically, in the mechanisms used to prevent overfitting in training, which is common in AI.
4) What methods did the authors use to validate the simulation results? They used a standard protocol for identifying cardiopulmonary sounds, and it is important to note how this was adapted for a real-time environment with IoT.
5) In addition, there are a few problems with the format of the manuscript, so the author should carefully modify it according to the standards of the journal.
Author Response
The paper is well written and documented. The organization is good. The presentation is good. Readability is good. References are regular. Although more detailed considerations on performance would improve the value of the paper, it seems to me suitable to be accepted for publication in this journal.
As noted in the comments below, there are a few issues that would benefit from clarification or further discussion before the paper is accepted:
1) Could you please rewrite the introduction? This section, in my opinion, is confusing to readers. I'd like to highlight a few key points: background, problem, proposed solution, and main contributions Furthermore, this would help to reinforce the topic's importance in the telehealth sector.
Response: As reviewer 1 pointed out, we revised the introduction section and added current references (Introduction section, line 26-43).
2) It is essential to include a separate section for the literature review, with the goal of comparing this work to others in the field and highlighting what makes it unique. It would also help to expand the references.
Response: As reviewer 1 pointed out, we added the literature review of remote auscultation systems using electronic stethoscopes in a new section (Discussion section 4.5, line 476-546).
3) The author should explain how to consider the interaction model to ensure the robustness of the architecture. Specifically, in the mechanisms used to prevent overfitting in training, which is common in AI.
Response: In this study, we did not use the interaction model or AI in simulator or online medical care program. Therefore, we thought there was no risk at overfitting.
4) What methods did the authors use to validate the simulation results? They used a standard protocol for identifying cardiopulmonary sounds, and it is important to note how this was adapted for a real-time environment with IoT.
Response: As reviewer 1 pointed out, we added the requirement for validation in the limitation subsection (Discussion section 4.3 Limitations, line 453).
5) In addition, there are a few problems with the format of the manuscript, so the author should carefully modify it according to the standards of the journal.
Response: As reviewer 1 pointed out, we modified the format of the manuscript.
Reviewer 2 Report
Interesting question and hypothesis
However, improvements should be made:
the language
incorporate current literature
Shorten the introduction and fill it with more recent data
The methodology is clear but should be clarified with inclusion and exclusion criteria
The results could be presented more clearly. The question is, which results are important and why?
The presentation of the results is good, but the language of the content should be revised here and the most important 3 to 4 results should also be discussed controversially. I think this is completely missing.
Author Response
Interesting question and hypothesis
However, improvements should be made:
the language
Response: Although the manuscript had already undergone proofreading in English, as per reviewer 2’s suggestion, we again proofread the manuscript with the certification of proofread.
incorporate current literature
Response: As reviewer 2 pointed out, we added more current literatures.
Shorten the introduction and fill it with more recent data
Response: As reviewer 2 pointed out, we revised the Introduction section by shortening it and adding more recent data (Introduction section, line 26-43).
The methodology is clear but should be clarified with inclusion and exclusion criteria
Response: As reviewer 2 pointed out, we have included the inclusion and exclusion criteria in the Materials and Methods section (Materials and Methods section, lines 180-184).
The results could be presented more clearly. The question is, which results are important and why?
Response: As reviewer 2 pointed out, the result of primary outcome (the total scores for correctly identified normal or abnormal cardiopulmonary sounds) was important because remote auscultation was used as the screening tool for cardiopulmonary diseases. We clarified this in the Discussion section (Discussion section, lines 390–395).
The presentation of the results is good, but the language of the content should be revised here and the most important 3 to 4 results should also be discussed controversially. I think this is completely missing.
Response: Although the manuscript already had undergone proofreading in English, as per reviewer 2’s suggestion, we again proofread the manuscript with the certification of proofread. We also clarified 3rd and 4th principal results in the Discussion section (Discussion section, lines 405–432).
Round 2
Reviewer 1 Report
The authors made the changes listed above. However, more references that support the scientific rigor of this study should be included in the Introduction, Strengths, and Comparisons with Our Prior Work sections to reinforce and guarantee the publication of this manuscript.
In addition, there are a few problems with the format of the manuscript, so the author should carefully modify it according to the standards of the journal (i.e., Table 4 illustrates a spacing anomaly (section 3.2.2), section 3 has a different space than the other sections, and section 2.4 “simulator” has an empty space).
Author Response
Dear Reviewer 1,
Thank you for your interest and valuable recommendations. Below you will find the response to each of your comments.
The authors made the changes listed above.
However, more references that support the scientific rigor of this study should be included in the Introduction, Strengths, and Comparisons with Our Prior Work sections to reinforce and guarantee the publication of this manuscript.
Response: As reviewer 1 pointed out, we added references.
In addition, there are a few problems with the format of the manuscript, so the author should carefully modify it according to the standards of the journal (i.e., Table 4 illustrates a spacing anomaly (section 3.2.2), section 3 has a different space than the other sections, and section 2.4 “simulator” has an empty space).
Response: As reviewer 1 pointed out, we modified the format.
Thank you again for your valuable time and recommendations. We believe these corrections have improved the quality of the manuscript and we hope it is worthy of publication.
Please contact us if any of our responses have not satisfied you.
Sincerely,
Taro Shimizu, MD, PhD, MPH, MBA
Dokkyo Medical University Hospital, Kitakobayashi 880, Shimotsuga, Japan
Telephone: + 81-282-87-2498; fax: + 81-282-87-2502
e-mail: shimizu7@dokkyomed.ac.jp